

# A seven-year surveillance of epidemiology of malaria reveals travel and gender are the key drivers of dispersion of drug resistant genotypes in Kenya

Moureen Maraka[1,2], Hoseah M. Akala[2], Asito S. Amolo[1], Dennis Juma[2], Duke Omariba[2], Agnes Cheruiyot[2], Benjamin Opot[2], Charles Okello Okudo[2], Edwin Mwakio[2], Gladys Chemwor[2], Jackline A. Juma[2], Raphael Okoth[2], Redemptah Yeda[2] and Ben Andagalu[2]

[1] School of Health Sciences, Jaramogi Oginga Odinga University of Science and Technology, Bondo, Siaya, Kenya
[2] Department of Emerging Infectious Diseases (DEID), United States Army Medical Research Directorate-Africa Kenya (USAMRD-A Kenya)/Kenya Medical Research Institute (KEMRI), Kisumu, Kisumu, Kenya

Corresponding authors
Moureen Maraka,
maraka204@gmail.com
Ben Andagalu,
Ben.Andagalu@usamru-k.org

## ABSTRACT

Malaria drug resistance is a global public health concern. Though parasite mutations have been associated with resistance, other factors could influence the resistance. A robust surveillance system is required to monitor and help contain the resistance. This study established the role of travel and gender in dispersion of chloroquine resistant genotypes in malaria epidemic zones in Kenya. A total of 1,776 individuals presenting with uncomplicated malaria at hospitals selected from four malaria transmission zones in Kenya between 2008 and 2014 were enrolled in a prospective surveillance study assessing the epidemiology of malaria drug resistance patterns. Demographic and clinical information per individual was obtained using a structured questionnaire. Further, 2 mL of blood was collected for malaria diagnosis, parasitemia quantification and molecular analysis. DNA extracted from dried blood spots collected from each of the individuals was genotyped for polymorphisms in *Plasmodium falciparum* chloroquine transporter gene (*Pfcrt* 76), *Plasmodium falciparum* multidrug resistant gene 1 (*Pfmdr*1 86 and *Pfmdr*1 184) regions that are putative drug resistance genes using both conventional polymerase chain reaction (PCR) and real-time PCR. The molecular and demographic data was analyzed using Stata version 13 (College Station, TX: StataCorp LP) while mapping of cases at the selected geographic zones was done in QGIS version 2.18. Chloroquine resistant (CQR) genotypes across gender revealed an association with chloroquine resistance by both univariate model ($p = 0.027$) and by multivariate model ($p = 0.025$), female as reference group in both models. Prior treatment with antimalarial drugs within the last 6 weeks before enrollment was associated with carriage of CQR genotype by multivariate model ($p = 0.034$). Further, a significant relationship was observed between travel and CQR carriage both by univariate model ($p = 0.001$) and multivariate model ($p = 0.002$). These findings suggest that gender and travel are significantly associated with chloroquine resistance. From a gender perspective, males are more likely to harbor resistant strains than females hence involved in strain dispersion. On the other hand, travel underscores the role of transport network in introducing spread of resistant genotypes, bringing in to focus the need to monitor

gene flow and establish strategies to minimize the introduction of resistance strains by controlling malaria among frequent transporters.

# INTRODUCTION

In 2006, the World Health organization (WHO) recommended artemisinin-based combination therapy (ACTs) use as first-line drug for treatment of uncomplicated malaria following widespread failure of sulfadoxine-pyrimethamine combination treatment. The strategy relied on the fast acting artemisinin derivatives rapidly to bring down the parasite biomass, relieves symptoms while the partner drug with longer half-life clears the residual parasites therefore cushioning the artemisinin derivative from emerging resistance (*Nosten & White, 2007*). The most widely used combinations globally include artemether-lumefantrine (AL, Coartem), dihydroartemisinin-piperaquine (DHAPPQ), artesunate-amodiaquine (ASAQ) and artesunate-mefloquine. These treatments alongside other intervention have contributed to about 37% decline in malaria cases and 60% reduction in mortality during the last fifteen years (*Dhiman, 2019*). However, the recent reports of resistance to artemisinin derivative of the ACTs in Southeast Asia (*Woodrow & White, 2017*) and its pattern of migration (*Shetty et al., 2019*) threaten this trajectory due to fears that the global dissemination of these strains would follow historical pattern of drug resistant strain dispersal. Africa parasite diversity studies show that these strains have not yet been detected in Africa (*Kamau et al., 2015*), but shows extant subpopulations of *P. falciparum* in sub-Saharan Africa (*Amambua-Ngwa et al., 2019*) suggestive of parallel emergence of ACT resistance in the region given the varying first-line drugs in across Africa. Notwithstanding the origin of the initial resistance, it is essential to understand patterns of the dispersal present-day characterized strains in natural infections at local level as a stop-gap for containment should ACT resistant strains emerge.

Non-synonymous mutations in the kelch propeller domain (K13-propeller) in *Plasmodium falciparum* have been associated with artemisinin resistance in samples from Southeast Asia (*Witkowski et al., 2013*). Further, the waning efficacy of ACTs has been partly associated with failure of the quinoline partners in the combination (*Borrmann et al., 2008*). Though most of these drugs have not been policy recommended, cross-resistance has been shown to occur owing to similarity in mechanisms of actions (*Veiga et al., 2016*). A case in point is the diminished mefloquine sensitivity in Kenya (*Eyase et al., 2013*) yet mefloquine had neither been first-line nor second-line treatment in the country. Single Nucleotide Polymorphisms (SNPs) in the *P. falciparum* multidrug resistance protein (*Pfmdr1*) and *P. falciparum* chloroquine resistance transporter (*Pfcrt*) genes confer resistance to a number of anti-malaria drugs. Chloroquine resistance has been associated to *Pfcrt* K76T SNP (*Mohammed et al., 2013*; *Zhao et al., 2019*). *Pfmdr1* 86Y and Y184 haplotypes have been shown to modulate chloroquine (CQ) resistance in the presence of *Pfcrt* 76T mutation

(*Venkatesan et al., 2014*). Mefloquine (MQ) and lumefantrine (LU) sensitivities are linked to *Pfmdr* 1 86Y point mutation. Additionally, emerging *Pfcrt* K76 allele carrying parasites with *Pfmdr* 1 N86 and 184F in the last decade have been linked to declining susceptibility to LU, part of the AL treatment that is recommended in Kenya (*Achieng et al., 2015*; *Eyase et al., 2013*; *Venkatesan et al., 2014*), and a reciprocal rise in parasite susceptibility to CQ and amodiaquine (ADQ) (*Achieng et al., 2015*; *Veiga et al., 2016*). Conversely, our previous studies showed that the prevalence of *Pfcrt* K76 and N86 increased from 6.4% in 1995–1996 to 93.2% in 2014 and 0.0% in 2002-2003 to 92.4% in 2014 respectively (*Achieng et al., 2015*), a period when the first and second-line malaria drugs for Kenya are AL and DHAPPQ, respectively. A study done in Zanzibar reported that treatment with AL selects for chloroquine-susceptible *Pfcrt* K76 allele while re-infecting parasites harbor *Pfmdr* 1 N86 and 184F alleles (*Sisowath et al., 2009*).

Development of resistance is often attributed to acquisition of survival advantage of parasites against drugs (*Hastings & Donnelly, 2005*). The rate and pattern of spread of initial resistant strain is often dependent on multiple confounding factors that should be monitored alongside drug resistant surveillance programs concurrently (*Mbugi et al., 2006*). The confounding factors includes initial prevalence of mutations, population movement between transmission regions among others (*Bloland, 2001*). Distribution, control and prevention of malaria has been reported to be influenced by human population movement (HPM) from areas with high transmission rates to malaria-free areas (*Cohen et al., 2012*). Similarly, HPM has been shown to contribute to spread of drug resistant parasite strains elsewhere (*Lynch & Roper, 2011*). Other factors like age, gender, location, history of past malaria infection and previous malaria treatment have been shown to affect malaria prevention and control. These factors could consequently be associated with the dispersal and distribution of chloroquine resistance.

To combat drug resistance, the WHO recommends a robust surveillance system to monitor and detect emergence of resistant genotypes for timely containment replace the yellow highlighted (*WHO, 2009*). In areas with documented success and declining disease burden, there are calls for initiation of pre-elimination/elimination phase in malaria control and management (*Whittaker, Dean & Chancellor, 2014*; *WHO, 2015*). However, elimination process is facing numerous challenges. A study done in Nigeria reported research, political will/funding, attitude/behavior change, conflicts, terrorism/migration, climate change, insecticide resistance, drug resistance and treatment as challenges facing malaria elimination (*Aribodor, Ugwuanyi & Aribodor, 2016*), highlighting the importance of understanding the malaria drug resistance patterns alongside demographics/clinical factors and parasite characteristics among symptomatic malaria cases as a prerequisite for entry in pre-elimination phase.

As countries move towards the elimination phase, enhanced surveillance systems are required to ensure that every infection is detected, treated and reported (*WHO, 2016a*). However, there is lack of sufficient information to feed such a trend of reduced malaria burden in Kenya. Most studies done in Kenya often dwell on one site mostly endemic regions, with a smaller sample size and no detailed information on demographics. Therefore, this study aimed at determining the role of gender and travel in dispersion of

chloroquine resistant genotypes among individuals with symptomatic malaria in epidemic zones, Kenya.

## MATERIALS & METHODS

### Ethics Statement, Study Protocol, Sites and Subjects

This study was approved by the Kenya Medical Research Institute (KEMRI), Walter Reed Army Institute of Research (WRAIR) and Human Research Protection office institutional review boards (protocol numbers: KEMRI #1330, WRAIR #1384 and HRPO Log #A-19306.3) respectively. Seven participating clinical centers were all Ministry of Health facilities located in various malaria epidemic regions in Kenya (Fig. 1); Lake endemic region (Kisumu East and Kisumu West/Kombewa District Hospitals) both situated in Kisumu County (0°14′60.00″N 34°54′59.99″E), Highland epidemic region had (Kericho District Hospitals and Kisii teaching/referral hospital) situated in Kericho County (0°22′2.3″S 35°16′52.72″E) and Kisii County (0°40′49.7352″S 34°46′37.4196″E) respectively, Coast endemic region (Malindi District Hospital) situated in Kilifi County (3°13′25″S 40°7′48″E) and Semiarid seasonal region (Isiolo District, Isiolo county (0°52′60.00″N 38°40′0.12″E) and Marigat Sub district Hospital, Baringo County (0°28′N 35°59′E). At each participating facility, training of medical staff, capacity building, and facility upgrades were provided by the Global Emerging Infections Surveillance (GEIS) Program, U.S. Department of Defense. Subjects attending outpatient clinics from 2008 to 2014, at least 6 months old and suspected of having non-complicated *P. falciparum* malaria were invited to participate. Written informed consent was obtained from adult subjects (≥18 years of age) or legal guardians for subjects <18 years of age. Infants weighing less than 5 kilograms were excluded.

### Sample Collection and Preparation

2 ml of blood was collected from eligible subjects who had tested positive by rapid diagnostic test (RDT; Parascreen H (Pan/*Pf*), Zephyr Biomedicals, Verna Goa, India) for *P. falciparum* malaria. Additionally, FTA filter paper (Whatman Inc., Bound Brook, New Jersey, USA) was used to collect three blood spots of about 100 μl each for *P. falciparum* DNA extraction and molecular analysis. Two blood films on glass slides were made for Giemsa staining at the laboratory for microscopic examination, to confirm RDT results and determine parasitemia. For discrepancies between RDT and microscopy, microscopy determined the final result. Collected specimens for molecular analysis were stored in −20 °C freezers at the respective hospitals temporarily, before being transported to the KEMRI/USAMRD-A Kenya research laboratory in Kisumu for long-term storage at −80 °C. The remainder ~1.5 mL of blood was cryopreserved in liquid nitrogen for future culture and in vitro susceptibility testing.

Subjects' demographic information, malaria history, symptoms for current infections and physical examination findings were recorded in case report forms (CRFs). Subjects were treated with oral artemether-lumefantrine (AL; Coartem) administered over three consecutive days, a standard of care for *P. falciparum* malaria in Kenya. The first dose was observed by the study team and remaining doses were self-administered at home.

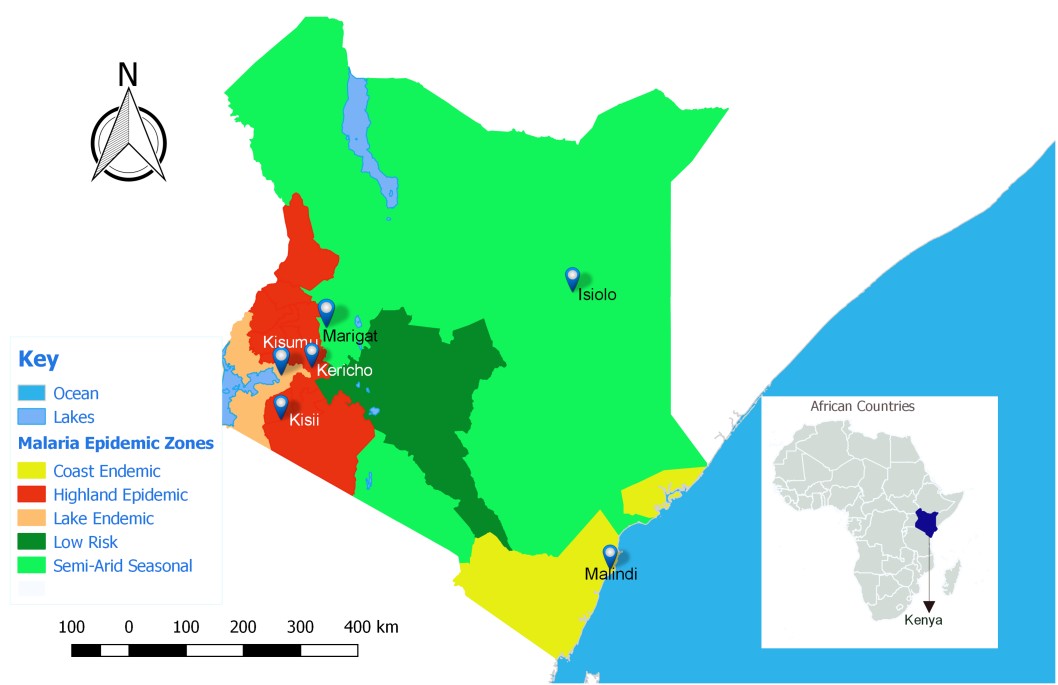

**Figure 1** **Map of Kenya showing the different malaria endemicity zones and locations of various surveillance hospitals.** Lake endemic region: Kisumu East and Kisumu West District Hospitals in Kisumu county, Semi-Arid, seasonal: Marigat District hospital in Baringo county and Isiolo District Hospital in Isiolo county. Coast endemic: Malindi District Hospital, Kilifi County. High land Epidemic: Kisii teaching and referral Hospital and Kericho District Hospital (QGIS version 2.18).

## Microscopy

Thick and thin smears were prepared for low and high parasitemia respectively. The smears were stained in fresh working Giemsa stain (diluted 1:10 in buffered water from stock) for 15–20 min and air dried. The parasite count was recorded as parasites per 200 white blood cells (WBCs) for thick smear and per 2,000 red blood cells for the thin smear. The two were standardized to parasites per microliter. An assumed WBCs count of 8,000/μl that has been accepted as reasonably accurate in estimating malaria parasite densities was used (*Afrane et al., 2014*; *Gyasi et al., 2015*). To convert parasite count per 2000 RBCs to parasites/μl an assumed RBCs count of $4.5 \times 10^6$/ μL was used (*Desai et al., 2015*).

## DNA extraction

DNA was extracted from blood spotted onto FTA filter papers using the QIAamp DNA Mini Kit (QIAGEN sciences, Maryland, USA) in accordance with the manufacturer's protocol for dried blood spots. The purified DNA was stored at −20 °C until required.

## *Pfcrt* K76T SNP analysis

A fragment of *Pfcrt* gene spanning K76T SNP was amplified by conventional polymerase chain reaction (PCR) on an Applied Biosystems' GenAmp PCR system 9700 (Foster City, CA, USA). The primer sequences used for the analysis are as described in *Wang et al. (2005)*. The primers were purchased from Applied Biosystems (Foster City, CA, USA).

The primary reaction was performed in a 15 µL reaction mixture which consisted of 0.5 µL AmpliTaq Gold (Applied Biosystems, Foster City, CA, USA), 2.5 µL of 10 µM forward primer and reverse primers, 2.5 µL of 10X PCR Buffer, 2.5 µL of 10 mM dNTPs, 2.0 µL of 25 Mm MgCl$_2$, 1 µL of DNA sample, and 4 µL water. The primary amplification was performed at the following conditions: 94 °C, 3 min for initial denaturation, 94 °C, 30 s for secondary denaturation and 45 cycles consisting of 56 °C, 30 s annealing, 60 °C for 30 s elongation followed by 64 °C for 3 min of final elongation and held at 4 °C. The primary PCR amplicon was then used as a template in the secondary PCR. The Secondary reaction was performed in a 25 µL reaction mixture which consisted of 0.5 µL AmpliTaq Gold (Applied Biosystems, Foster City, CA, USA), 2.5 µL of 1 µM forward primer and reverse primers, 2.5 µL of 10X PCR Buffer, 2.5 µL of 10 mM dNTPs, 1.5 µL of 25 mM MgCl$_2$, 3.7 µL of primary PCR product, and water 11.8 µL. The secondary amplification was performed at the following conditions: 94 °C, 3 min for initial denaturation, 94 °C, 30 s for secondary denaturation and 20 cycles consisting of 47 °C, 30 s annealing, 64 °C for 30 s elongation followed by 64 °C for 3 min of final elongation and held at 4 °C. The samples were run in duplicate with one set containing primers targeting the wild type allele (K76) while the second had primers specific for the mutant allele (76T). 5 µL of the amplicon was mixed with 5 µL of gel loading dye on a parafilm and run concurrently with reference DNA ladder on a 2% gel. The two products per sample from secondary PCR were run concurrently on agarose gel to discriminate between the wild type, mutant and mixed alleles. Ethidium bromide dye was used to stain the agarose gel for visualization under UV light. The electrophoresis was performed at 97 volts for 30 min for both primary and secondary products. The bands on the gels were visualized immediately under UV light system (Ultra Violet Products Inc. Transilluminator). DNA from W2 and D6 *Plasmodium falciparum* clones were used as positive controls for mutant and wild type respectively while distilled water was used as a negative control. The negative control was not expected to show any band on gel after electrophoresis while positive control and positive samples were expected to have sizes of 560 base pairs (bps) for primary product, 342 bps for both mutant and wild type of the secondary product.

### *Pfmdr*1 N86Y and *Pfmdr*1 Y184F SNP analysis

A fragment of *Pfmdr*1 gene spanning N86Y and Y184F SNPs was amplified to determine mutations by real-time PCR machine, Applied Biosystems' prism 7500 Fast real-time PCR system (Foster City, CA, USA). The primer and probe sequences used are described elsewhere (*Purfield et al., 2004*). The probes were labeled with 6FAM (for wild type) or VIC (for Mutant), at their 5′ ends and non-fluorescent quencher (TAMRA) covalently attached at the 3′ end for each. Each well on PCR plate contained both the wild type and mutant type primers and probes. The presence of either or both alleles was showed from the increase of the FAM or VIC fluorophores in distinguishing wild, mutant or mixed alleles. These primers and probes were purchased from Applied Biosystems (Foster City, CA, USA). The reactions were performed in a 25 µL reaction mixture made of 18.75 µL of the master mix and 6.25 µL of the *Plasmodium* DNA. The master mix consisted; 6.55 µL of water, 5 µL of 25 mM MgCl$_2$, 20 µL of Amplitaq Gold, 2.5 µL of 10X PCR buffer, 2.5 µL of 100 Mm, 0.5

µL of 100 µM of *Pfmdr1* (86 or 184,) forward primer, 0.5 µL of 100 µM of *Pfmdr1* (86 or 184) reverse primer, 0.25 µL of 100 µM of *Pfmdr1* (86 or 184) wild type probe and 0.25 µL of 100 µM of *Pfmdr1* (86 or 184) mutant probe. Amplification was performed at the following conditions: 1 cycle at 50 °C for 1 min, 1 cycle at 95 °C for 10 min and 45 cycles consisting of 95 °C for 15 s followed by 58 °C for 1 min.

Quality control was done by running samples and controls in triplicates. Additionally, quality control amplification results were valid only when the positive control (clones W2-mutant and D6-wild type) generated the acceptable average cycle threshold ($C_T$) value $\leq 30$ while the negative control (distilled water) remained unamplified therefore undetected.

### Data management and analysis

Numerical data were expressed as proportions and compared using Pearson's Chi-square. Parasitemia expressed as parasites/microliter were log transformed to the natural log and presented as medians with inter quartile range (IQR) and range. Comparison of parasitemia across age and regions was determined by Kruskal–Wallis test by ranks or one way analysis of variance (ANOVA) on ranks. Dunn's pairwise multiple comparison test was used for post hoc analysis. Univariate and multivariate Logistic regression was used to assess the association of demographic and clinical factors to chloroquine resistance molecular markers. Significance levels were set at 0.05 at 95% confidence interval. Statistical analyses were performed using Stata version 13 (Stata Corp LP, College Station, Texas) while mapping was done in QGIS version 2.18.

## RESULTS

A total of 1,776 subjects presenting with uncomplicated malaria from four malaria epidemic zones across Kenya between 2008 and 2014 were evaluated. The lake endemic zone enrolled the highest number of subjects owing to high disease endemicity followed by highland epidemic zone while Semi-arid seasonal had the least. The demographic and clinical presentation characteristics are summarized in Table 1. Both genders were equally represented with 48.8% ($n = 867$) male and 51.2% ($n = 909$) female. Majority 56.7% ($n = 999$) were children below five years followed by 6–15 years 21.9% ($n = 386$) and above 16 years 21.3% ($n = 376$).

All subjects enrolled by this study had a mean temperature of 38.1 $\pm1.21$, mean $\pm$ standard deviation (SD). The chief complaint was fever followed by headache that is in line with classical malaria presentation. At least 62.4% ($n = 1,109$) subjects had not travelled from their residence within two months prior to enrolment to the study. 80.2% ($n = 1,422$) reported to have had at least one episode of malaria before date of enrollment. Of these, 10.6% ($n = 188$) had been treated for malaria in the last 6 weeks prior to enrollment.

### Malaria parasite density

The log transformed parasitemia for the 1,776 samples assessed as number of parasites diagnosed by microscopy per microliter of whole blood was median 10.4, IQR 9.1–11.5

**Table 1 Demographic and clinical characteristics of subjects enrolled by resistance.**

| Characteristics | Malaria Resistant Status | | | | |
| --- | --- | --- | --- | --- | --- |
| | Resistant n (%) $n = 941$ | Non-Resistant n (%) $n = 835$ | Total n (%) $n = 1,776$ | Chi Square | p - value |
| **Sex** | | | | 1.44 | 0.23 |
| Male | 472(54.4%) | 395(45.6%) | 867(48.8%) | | |
| Female | 469(51.6%) | 440(48.4%) | 909(51.2%) | | |
| **Age group, years** | | | | 0.4 | 0.82 |
| ≤5 | 520(52.1%) | 479(47.9%) | 999(56.7%) | | |
| 6–15 | 206(53.4%) | 180(46.6%) | 386(21.9%) | | |
| ≥16 | 202(53.7%) | 174(46.3%) | 376(21.3%) | | |
| **Mean body temperature ±SD** | 38.1 ± 1.24 | 38.1 ± 1.18 | 38.1 ± 1.21 | ** | ** |
| **Ever travelled in the last 2 months** | | | | 2.36 | 0.125 |
| Yes | 368(55.3%) | 297(44.7%) | 665(37.5%) | | |
| No | 572(51.6%) | 537(48.4%) | 1,109(62.5%) | | |
| **Ever had malaria** | | | | 0.79 | 0.372 |
| Yes | 746(52.5%) | 676(47.5%) | 1,422(80.2%) | | |
| No | 194(55.1%) | 158(44.9%) | 352(19.8%) | | |
| **Treated for malaria in the last 6 weeks** | | | | 8.07 | 0.004 |
| Yes | 118(62.8%) | 70(37.8%) | 188(10.6%) | | |
| No | 822(51.8%) | 764(48.2%) | 1,586(89.4%) | | |
| **Chief Complaint** | | | | 0.46 | 0.978 |
| Fever | 611(52.5%) | 552(47.5%) | 1,163(66.2%) | | |
| Headache | 200(54.1%) | 170(45.9%) | 370(21.1%) | | |
| Coughing | 10(55.6%) | 8(44.4%) | 18(1.0%) | | |
| Joint pains | 20(50.0%) | 20(50.0%) | 40(2.3%) | | |
| Others | 87(52.4%) | 79(47.6%) | 166(9.4%) | | |

**Notes.**
** Not applicable for chi-square test. *P* value < 0.05 was considered significant.
SD, standard deviation.

and range 3.7–14.2. Using Kruskal–Wallis test by ranks, there was a significant difference [F (3) = 282.4, $p < 0.0001$] in parasitemia across the seven study sites located in the four malaria endemicity zones of Kenya. The lake endemic region had the highest parasitemia median 10.8 IQR 9.9–11.6 range 3.7–14.2 followed by coast endemic region median 9.9 IQR 8.8–10.7 range 4.8–11.7 while highland epidemic had the least median 8.9, IQR 7.7–10.4, range 4.4–14.2 (Fig. 2). Post hoc analysis by Dunn's test showed that the lake endemic region has a significantly higher parasitemia than the rest of the transmission regions ($p < 0.05$) as shown in Fig. 2.

When comparing parasite burden across age groups, individuals in age group <5 years had marginally higher parasitemia median 10.6, IQR 9.3–11.6 range 3.7–13.9 followed by that of age group 6–15 years, median 10.4, IQR 9.2–11.4 range 4.4–14.2. There was a significant difference [F (2) = 54.94, $p < 0.0001$] in parasite density among the age groups. Age group ≥ 16 years parasitemia differed significantly from those of both age groups of <5 years and 6–15 years ($p < 0.05$) (Fig. 3).

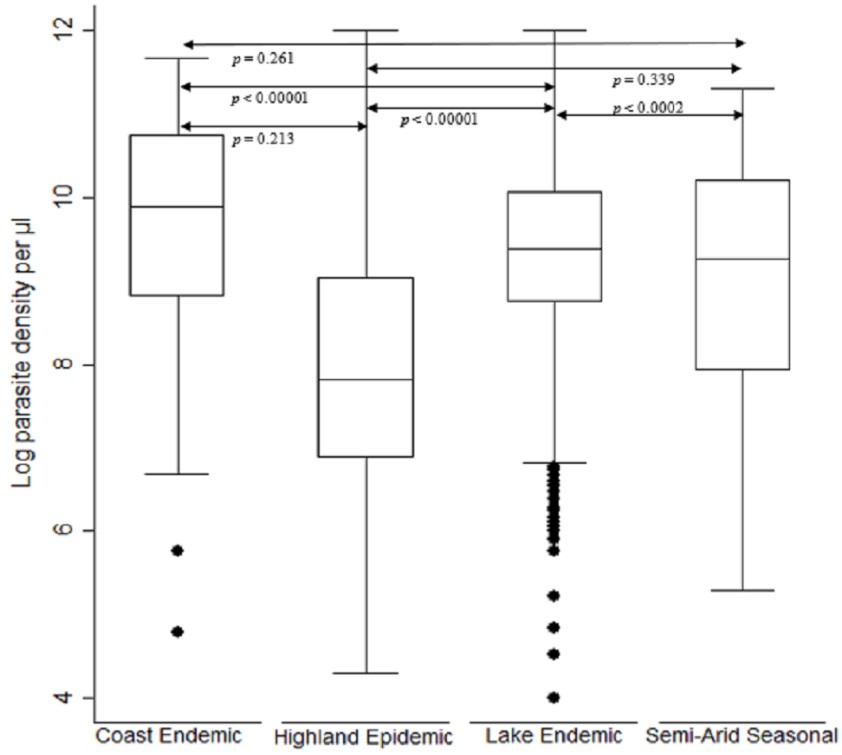

**Figure 2 Comparison of parasite density by malaria epidemic zones.** Parasite density (natural log transformation). Dots represent outliers. *P* value < 0.05 was considered significant.

## Trends of *Pfcrt* 76, *Pfmdr*1 86 and *Pfmdr*1 184 polymorphisms

We established trends and distribution of chloroquine resistant genotypes; *Pfcrt* 76, *Pfmdr* 1 86 and *Pfmdr* 1 184 across Kenya between 2008 and 2014. The average frequency of the mutant genotype, *Pfcrt* 76T was 28.3% ($n = 446$), Wild-type *Pfcrt* K76 was 49.4% ($n = 780$) while mixed infection comprising both wild-type and mutant accounted for 22.3% ($n = 353$) infections countrywide (Table 2). Notably, there was a decline in frequency of *Pfcrt* 76T single nucleotide polymorphism (SNP) between 2008 and 2014 by 66% ($n = 125$) to 5.5% ($n = 27$) respectively (Fig. 4). There was a significant difference among the *Pfcrt* 76 polymorphisms across the years by Chi-square test [$\chi^2(6) = 39.55$, *p <0.0001*].

The distribution by the current years in the study (2012 to 2014) mutant genotype *Pfcrt* 76T by malaria endemicity zones showed that Coast endemic region had the highest 34.8% ($n = 8$) frequency as compared to highland epidemic 11.2% ($n = 36$), Lake endemic 26.2% ($n = 39$) and semi-arid seasonal with 5.0% ($n = 3$) as shown in Fig. 5. There was a significant difference in frequency of *Pfcrt* 76T polymorphism across the zones ($p < 0.0001$) over the study period. Distribution by age was 27.6% ($n = 243$), 26.9% ($n = 94$), 29.9% ($n = 100$) for ages ≤5, 6–15, ≥16 years respectively during the study period. There was no difference in *Pfcrt* 76T frequency across the age groups.

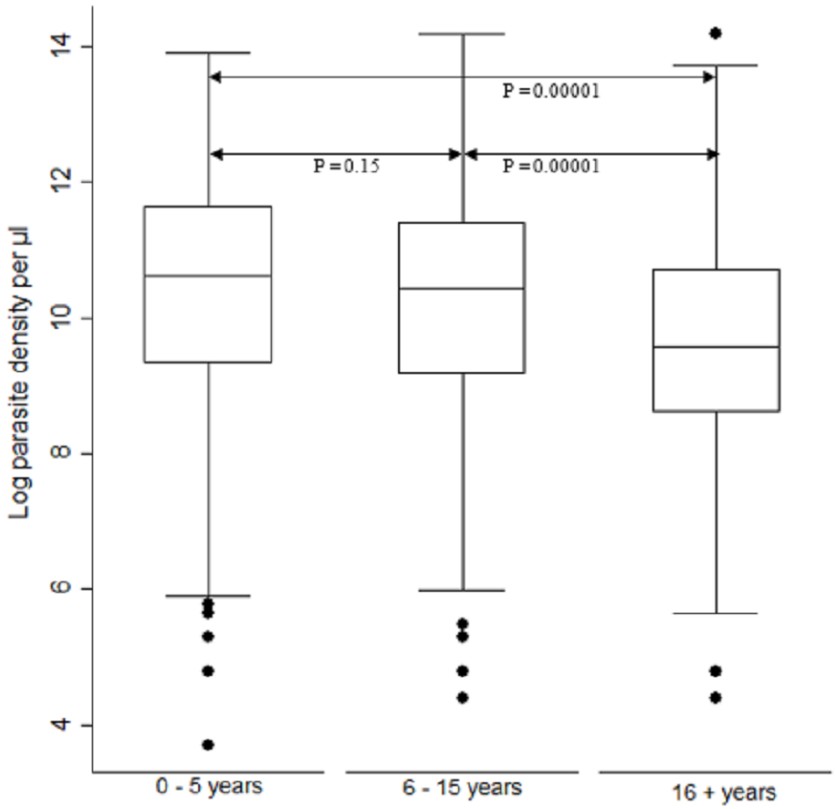

**Figure 3 Comparison of parasite densities by age groups.** Parasite density (natural log transformation). Dots represent outliers. *P* value < 0.05 was considered significant.

The average frequency of the mutant genotype, *Pfmdr* 1 184F was 32.3% ($n = 507$), wild type *Pfmdr* 1 Y184 was 47.1% ($n = 739$) while mixed infection accounted for 20.6% ($n = 324$) infections countrywide (Table 2). A three-fold increase in *Pfmdr* 1 184F, SNP was noted between 2008 and 2014 with frequency of 13.7% ($n = 27$) to 37.2% ($n = 191$) (Fig. 4). There was a significant difference among the *Pfmdr* 1 Y184F polymorphisms across the years [$\chi^2(6) = 17.07$, *p <0.009*].

The distribution by the current years in the study (2012 to 2014) of mutant genotype *Pfmdr* 1 184F by malaria transmission zones showed a marginally higher frequency in the Semi-arid seasonal 45.0% ($n = 9$) followed by Lake endemic 38.4% ($n = 161$), Highland epidemic 36.8% ($n = 118$) and Coast endemic 13.0% ($n = 3$) as shown in Fig. 5. There was a significant difference in frequency of *Pfmdr* 1 184F SNP among the zones ($\chi^2 = 17.1$, *p <0.009*), (Table 2) across the study period. The *Pfmdr* 1 184F, SNP was comparable across all age groups 31.4% ($n = 280$) during the study period, 34.3% ($n = 118$), 32.9% ($n = 105$) for ages ≤5, 6–15, ≥16 years respectively. There was no difference in *Pfmdr* 1 184F genotype mutation across the age groups.

The average frequency of the mutant genotype, *Pfmdr* 1 86Y was 13.7% ($n = 229$), wild type *Pfmdr* 1 N86 was 73.9% ($n = 1,238$) while mixed infection comprising both wild-type and mutant accounted for 12.5% ($n = 209$) infections countrywide (Table 2). Notably,

**Table 2  Distribution of *Pfcrt* 76, *Pfmdr1* 86 and 184 single nucleotide polymorphisms by zone.**

| Polymorphisms (n = 1,676) | Lake endemic | Highland epidemic | Semi-Arid seasonal | Coast endemic | Total | Chi square | p - value |
|---|---|---|---|---|---|---|---|
| *Pfcrt* −76 | | | | | | | |
| Mutant | 302(29.6%) | 129(26.2%) | 3(11.1%) | 12(31.6%) | 446(28.3%) | 39.5 | 0.0001 |
| Wild-type | 467(45.7%) | 282(57.2%) | 21(77.8%) | 10(26.3%) | 780(49.4%) | | |
| Wild-type/mutant | 252(24.7%) | 82(16.6%) | 3(1.1%) | 16(42.1%) | 353(22.3%) | | |
| *Pfmdr* 1-86 | | | | | | | |
| Mutant | 155(14.4%) | 56(10.8%) | 3(12.5%) | 15(27.3%) | 229(13.7%) | 21.9 | 0.001 |
| Wild-type | 769(71.5%) | 415(79.7%) | 19(79.2%) | 35(63.6%) | 1,238(73.9%) | | |
| Wild-type/mutant | 152(14.1%) | 50(9.6%) | 2(8.3%) | 5(9.1%) | 209(12.5%) | | |
| *Pfmdr* 1 184 | | | | | | | |
| Mutant | 319(33.1%) | 168(32.3%) | 13(32.5%) | 7(13.5%) | 507(32.3%) | 17.1 | 0.009 |
| Wild-type | 434(45.0%) | 255(49.0%) | 13(32.5%) | 37(71.2%) | 739(47.1%) | | |
| Wild-type/mutant | 212(21.9%) | 97(18.7%) | 7(17.5%) | 8(15.4%) | 324(20.6%) | | |

**Notes.**

Samples with mutant, wild-type and a mixture of mutant/wild-type were identified for each epidemic zone. Undetermined samples contained neither mutant, Wild-type nor a mixture of mutant/Wild-type.

$P < 0.05$ was considered significant.

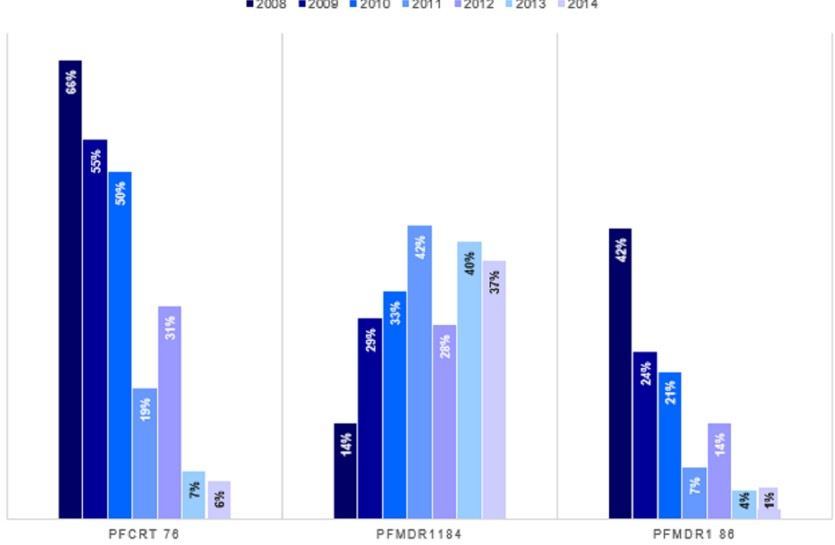

**Figure 4   Trends of genotype frequencies of *Pfcrt* 76T, *Pfmdr* 1 184F and *Pfmdr* 1 86Y polymorphisms between 2008 and 2014.** There was a significant difference ($p < 0.0001$) for each of the *Pfcrt* 76T, *Pfmdr* 1 184F and *Pfmdr* 1 86Y SNPs with $\chi^2$ of 354, 780, 103 respectively from 2008 to 2014. *Pfcrt* 76 ($n = 1,579$), *Pfmdr* 1 86 ($n = 1,676$) and *Pfmdr* 1 184 (1570).

there was a decline of the *Pfmdr* 1 86Y SNP from 41.8% ($n = 81$) to 1.4% ($n = 7$) between 2008 and 2014 countrywide (Fig. 4). There was a significant difference among the *Pfmdr* 1 N86Y polymorphism across the years [$\chi^2(6) = 21.87, p < 0.001$].

The distribution by current years in the study (2012 to 2014) of mutant genotype, *Pfmdr* 1 86Y by malaria transmission zones showed that the coastal endemic had the highest
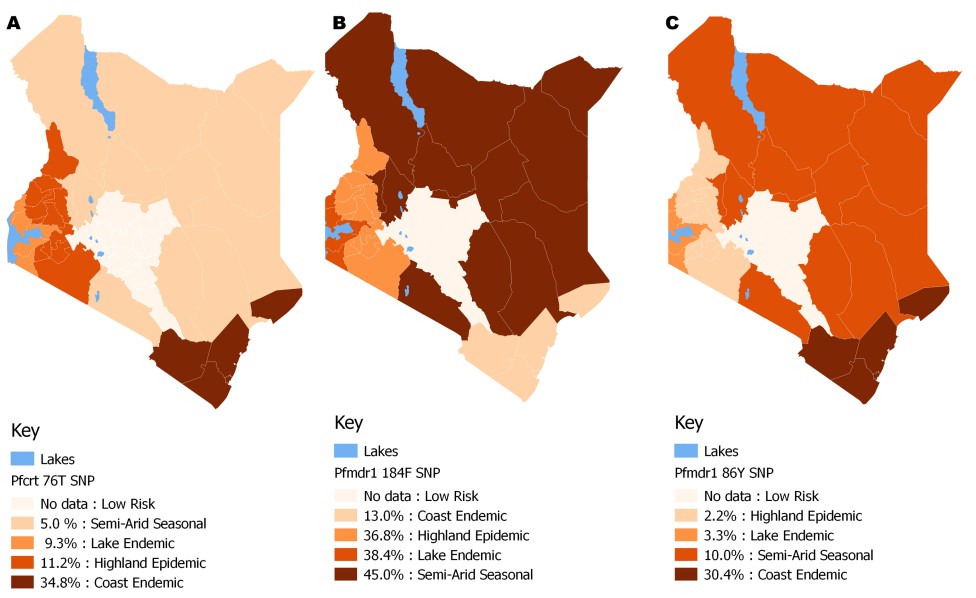

**Figure 5** **Spatial distribution patterns of (A)** *Pfcrt* **76T, (B)** *Pfmdr* **1 184F and (C)** *Pfmdr* **1 86Y SNPs by malaria epidemic zones using QGIS version 2.18.** There was a significant difference among the *Pfcrt* 76T, *Pfmdr1* 184F and *Pfmdr1* 86Y SNPs across the epidemic zones ($\chi 2$ (6) = 39.5, $p < 0.0001$), ($\chi 2$ (6) = 21.9, $p < 0.001$) and ($\chi 2$ (6) = 17.1, $p < 0.009$) respectively.

30.4% ($n = 7$) frequency followed by Semi-arid seasonal 10% ($n = 2$), Lake endemic 3.3% ($n = 14$) and Highland epidemic 2.2% ($n = 7$) (Fig. 5). There was a significant difference in frequency of *Pfmdr* 1 86Y SNP (Table 2) across the study period and was comparable among all age groups 12.8% ($n = 122$), 13.4% ($n = 49$), 16.3% ($n = 56$) for ages $\leq 5$, 6-15, $\geq 16$ years respectively during the study period. There was no difference in *Pfmdr* 1 86 genotype mutation across the age groups.

## Demographic and clinical factors associated with chloroquine resistance

To explore the association between demographic factors and chloroquine resistance (CQR) based on *Pfcrt* 76 allele, a CQR primary marker, univariate and multivariate logistic regression was used (Table 3). Demographic factors included sex, age group, and travel in the last 2 months, ever had malaria, treated for malaria within the last 6 weeks and the malaria transmission zones. Sex was significantly associated with CQR, *p <0.027* by univariate model and *p <0.025* by multivariate model. Males were more likely to have CQR, 1.28 times by univariate model and 1.29 times by multivariate model. On the other hand, age was not significantly associated with CQR, though it's important to note that subjects above 6 years of age were about one more time likely to harbor resistant genotypes. There was a significant relationship between having travelled in the last two months and CQR both by univariate model ($p < 0.001$) and multivariate model ($p < 0.002$), with subjects who had travelled being one or more time likely to be chloroquine resistant than those who had not. Having had malaria infection prior to the present visit was not associated with

**Table 3  Association of demographic and clinical factors with CQR.**

| Parameter | Chloroquine | | | Univariate | | Multivariate | |
|---|---|---|---|---|---|---|---|
| | Resistant (n = 446) | Non-Resistant (n = 1,330) | Total (n = 1,776) | OR(95% CI) | p- value | OR(95% CI) | p- value |
| **Sex** | | | | | | | |
| Female | 208(22.8%) | 701(77.1%) | 909(51.2%) | ref | | ref | |
| Male | 238(27.4%) | 629(72.6%) | 867(48.8%) | 1.28(1.03–1.58) | 0.027 | 1.29(1.03–1.61) | 0.025 |
| **Age group, years** | | | | | | | |
| ≤5 | 243(24.3%) | 756(75.7%) | 999(56.7%) | ref | | ref | |
| 6–15 | 94(24.4%) | 292(75.7%) | 386(21.9%) | 1.00(0.76–1.32) | 0.991 | 1.08(0.82–1.43) | 0.577 |
| ≤16 | 100(26.6%) | 276(73.4%) | 376(21.4%) | 1.13(0.86–1.48) | 0.386 | 1.20(0.90–1.59) | 0.207 |
| **Ever travelled in the last 2 months** | | | | | | | |
| Yes | 198(29.7%) | 467(70.2%) | 665(37.5%) | 1.47(1.18–1.83) | 0.001 | 1.44(1.14–1.79) | 0.002 |
| No | 248(22.4%) | 861(77.6%) | 1109(62.5%) | ref | | ref | |
| **Ever had malaria** | | | | | | | |
| Yes | 354(24.9%) | 1068(75.1%) | 1422(80.2%) | 0.94(0.72–1.22) | 0.631 | 0.81(0.60–1.08) | 0.155 |
| No | 92(26.1%) | 260(74.6%) | 352(19.8%) | ref | | ref | |
| **Treated for malaria in the last 6 weeks** | | | | | | | |
| Yes | 58(30.9%) | 130(69.1%) | 188(10.6%) | 1.38(0.99–1.92) | 0.056 | 1.46(1.03–2.06 | 0.034 |
| No | 388(24.5%) | 1198(75.5%) | 1586(89.4%) | ref | | ref | |
| **Epidemic Zones** | | | | | | | |
| Coast Endemic | 12(19.4%) | 52(80.7%) | 62(3.5%) | ref | | ref | |
| Highland Epidemic | 129(23.3%) | 424(76.7%) | 553(31.1%) | 1.27(0.65–2.45) | 0.027 | 1.44(0.72–2.9) | 0.302 |
| Lake Endemic | 302(26.8%) | 823(73.2%) | 1125(63.3%) | 1.52(0.8–2.91) | 0.184 | 1.87(0.93–3.74) | 0.077 |
| Semi-Arid Seasonal | 3(8.3%) | 33(91.7%) | 36(2.0%) | 0.38(0.09–1.45) | 0.124 | 0.38(.09–1.52) | 0.174 |

**Notes.**

Ref, Reference group; OR, Odds ratio; CI, confidence interval.

$P < 0.05$ was considered significant.

CQR, but individuals who had suffered from malaria before were one more time likely to be CQR than those who had no malaria infection prior to study enrollment. On the contrary, treatment in the last 6 weeks before visit was significantly associated with CQR by multivariate model ($p < 0.034$). Individuals who had treatment six weeks prior to the visit were one more time likely to acquire CQR. No association between living in malaria epidemic zones and CQR was observed. However, individuals in Lake endemic region are more likely to be CQR by 1.52 times (univariate model) and 1.87 times (multivariate model). Lake endemic region was followed by highland epidemic while people living in semiarid seasonal were less likely to be CQR.

## DISCUSSION

Antimalarial resistance has significantly hindered malaria control efforts. Since effective vaccines for malaria are still way off, and drugs are most relied upon, success of malaria control strategies are dependent on understanding factors that influence dispersion of malaria drug resistance genotypes. Most studies on CQR appear inconclusive on methods containing it's spread due to insufficient demographic and clinical information (*Akala et*

*al., 2014*; *Angira et al., 2010*; *Zhong et al., 2008*). This study showed that travel, gender and clinical factors were associated with chloroquine resistance among symptomatic malaria cases in four of the five malaria epidemic zones in Kenya. Additionally, this study established that while *Pfcrt* 76 and *Pfmdr* 1 86 SNPs had reduced, the *Pfmdr* 1 184 SNP was on the rise.

Travel and gender were significantly associated with CQR among symptomatic malaria individuals by both univariate and multivariate model (Table 3). The association between gender and CQR implies that males are more likely to be chloroquine resistant than females, though literature indicates that gender norms exposes females and children to the risk of malaria (*Ministry of Health K, 2015*). On the contrary, recent studies in Yemen and Malaysia did not show association between CQR and gender (*Atroosh et al., 2012*; *Bamaga, Mahdy & Lim, 2015*) despite compelling evidence of potential role of gender in epidemiology of malaria elsewhere (*Diiro et al., 2016*; *WHO, 2007*). The contrary findings could be attributed to the social cultural values around gender. A study done on prevalence of asymptomatic and submicroscopic malaria infections on Islands in Lake Victoria, Kenya, reported to have a significant prevalence of malaria in males than in females ($p < 0.005$) (*Idris et al., 2016*). Globally, gender dimensions also influence access to treatment, care for malaria and use of preventive measures (*WHO, 2007*). Females play the primary role of care giving to other members in the household, including leading the majority of health care seeking role for the rest of the family members (*Diiro et al., 2016*; *Roll back malaria partnership, 2016*) though males still dominate decision making on health and economic issues in the households (*Diiro et al., 2016*). Malaria is "gender blind" as mosquitoes bite indiscriminately (*Ministry of Health K, 2015*). However, in endemic regions, males may delay seeking treatment and risk accumulating super infection. Research has shown that mutant strains that develop drug resistance enhance virulence (*Shahinas, Folefoc & Pillai, 2013*) wanton utilization of resources and therefore eliciting symptoms that distorts the host/parasite balance that would allow wild type parasites to remain asymptomatic (*Akala et al., 2014*; *Deroost et al., 2016*).

The current finding on association of travel and CQR is concurrent with similar studies reported. A study done in Ethiopia reported effects of migration on parasite gene flow promoting transmission of chloroquine resistance (*Lo et al., 2017*). At a smaller scale, another study in Congo, reported evidence of imported resistant *P. falciparum* strains into Guatemala from Congo given globalization and international travels. Travelers from endemic regions with malaria symptoms should be suspected of harboring chloroquine resistant strains (*Patel et al., 2014*). Human population movement has been shown to contribute to spread of drug resistant parasite strains elsewhere (*Lynch & Roper, 2011*).

It was also noteworthy that treatment in the last 6 weeks before enrollment was not associated with CQR by univariate model but was associated by multivariate model. This is an implication that other factors contribute to CQR. Literature shows that irrational use of drugs (*Nsubuga et al., 2011*) gender dimensions (*WHO, 2007*), travel (*Juliao et al., 2013*), number of parasites exposed to a drug, the drug concentration to which the parasites are exposed, and the simultaneous presence of other antimalarials in the blood to which the parasite is not resistant influence the emergence and spread of drug resistant malaria parasites (*WHO, 2018*). These multiple factors argue for deployment of integrated research

for effective malaria control. Such a strategy would include systems thinking, out-come-oriented monitoring and evaluation approaches, collaboration - across disciplines, sectors and regions -, multi-stakeholder engagement, and sensitivity to social equity (*Wiese, 2012*).

Other factors such as age, ever had malaria and location by epidemic zone were not associated with CQR. However, age has been shown to be a risk factor of malaria with a vast majority of malaria cases occurring under age of 5 years (*WHO, 2016b*). Though children who live in endemic areas may have parasites with drug resistance to *Plasmodium falciparum*, they often recover after chemotherapy. This is an indication that acquired immunity works in synergy with antimalarials (*Enevold et al., 2007*). On the other hand, the history of malaria is important in determining recurrent malaria, recrudescence of malaria, relapse and infection (*Velho et al., 2014*). Recurrence has been reported to be due to one or many of the following: therapeutic failure resulting from non-adherence to treatment, resistance of the parasite to the drug-use, poor quality of the medication, or sub-therapeutic doses of the drugs; (b) reactivation of hypnozoites; and (c) exposure to new infection by the mosquito vector (*Hedt, Laufer & Cohen, 2011*; *White, 2011*). Additionally, though location by epidemic zones was also not associated with CQR, determining location of origin of patients who present with malaria is key for successful intervention strategies (*Pindolia et al., 2013*).

High parasitemia was noted for samples from across all sites (Fig. 2) and age groups (Fig. 3) in Kenya regardless of the difference in disease transmission burden. Studies suggest that residents of holoendemic regions are often immune to malaria (*Rolfes et al., 2012*). They therefore can harbor parasitemia for long prior to showing symptoms than those in non-immune individuals residing in highland epidemic and semi-arid seasonal transmission zones of Kenya. It is however worth noting that microscopy that is the Kenya ministry of health's guidelines recommended diagnosis method is less sensitive for detection of low parasitemia (*Lo et al., 2015*), a factor that could have excluded individuals with low parasitemia. Additionally, a study done in Tanzania reported a two-fold difference in parasite density when samples were collected at two time points ($-2$ and 0 h) prior to treatment (*Carlsson et al., 2011*), It is therefore possible that parasitemia for low transmission region would be higher than that reported here if all sub-microscopic parasitemia's were depicted using sensitive methods and if two samples were collected at different time points. Further, the study design that included only symptomatic individuals with detectable malaria could have been biased towards individuals with higher load across all sites though similar studies in Southeast Asia have reported significantly lower parasitemia (*Chaorattanakawee et al., 2013*).

Analysis of trends and patterns of distribution of chloroquine resistant genotypes revealed a notable decline by 56.7% in the *Pfcrt* 76T and 38.8% for the *Pfmdr* 1 86Y mutant genotypes between 2008 and 2014 (Fig. 4). This presents a possibility for future introduction of chloroquine use for malaria treatment as a combination therapy. On the other hand, this observation indicates an increased risk of reinfection after treatment with AL. A study done to determine effects of *Pfcrt* and *Pfmdr* 1 gene polymorphisms on therapeutic responses to artesunate-amodiaquine (ASAQ) and AL reported that parasites with the *Pfmdr* 1 N86, and *Pfcrt* K76 alleles re-infected patients earlier after AL treatment (Venkatesanal et al., 2014),
due to selection of less susceptible parasites (*Sisowath et al., 2009*). Another study done on genetically engineered *P. falciparum* lines reported that *Pfmdr1* N86Y mutation increased parasite susceptibility to a wide range of first line antimalarials including Lumefantrine and Dihydroartemisinin (DHA) while decreasing parasite susceptibility to CQ. *Pfmdr1* N86Y mutation was reported to alter parasite response DHA when compared to N86 parasites (*Veiga et al., 2016*). The mixed genotype *Pfcrt* K76T mutant and wild type allele is directly linked to both in vitro and clinical resistance. Therefore, it can still be used as a biomarker for CQ resistance (*Wellems & Plowe, 2001*).

On the other hand, there was a reciprocal increase for the *Pfmdr* 1 184F mutant genotype from 13.7% to 37.2% representing a 23.5% rise between 2008 and 2014 (Fig. 4). The implication of *Pfmdr* 1 Y184F mutation has been reported to be minimal on antimalarial drug susceptibilities (*Veiga et al., 2016*). This observation was contrary to a study reported in Grande Comore Island, Comoros between 2006–2014 where *Pfmdr* 1 184F mutations dropped from 52.2 to 30.0% ($p < 0.01$) (*Huang et al., 2016*). Another study done in Guinea-Bissau between 2003 and 2012 reported that there was no significant change in the *Pfmdr* 1 184F mutant allele (*Jovel et al., 2015*). A study done in Yemen, Tehama region reported a high prevalence of *Pfmdr* 1 184F mutant allele (99%) (*Atroosh et al., 2016*)and an increase from 19.6 to 22.7% in 2010–2012 in Maputo after introduction of ACT (*Lobo et al., 2014*). In Kenya, a significant association between polymorphisms at the *Pfmdr* 1 184 and lumefantrine have been reported (*Achieng et al., 2015*). However, a previous study from Odisha has reported the high prevalence of *Pfmdr* 1 86Y mutation and low prevalence of *Pfmdr* 1 184F mutation (*Antony et al., 2016*). Additionally, increased *Pfmdr1* copy number has been associated with treatment failure after short 4-dose AL therapy (*Sisowath et al., 2009*). The increased *Pfmdr1* copy numbers has been reported to be a rare observation in African population due to little exposure to MQ (*Venkatesan et al., 2014*).

Significant variation in genotype distribution was noted; the Coast and Lake endemic regions had the highest frequency of 31.6% and 29.6% for the *Pfcrt* 76T mutant allele respectively (Table 2). Genotype distribution for year 2014 (Fig. 5) by spatial analysis also showed varied distribution (Fig. 5). This observation could be partly due to inadequate enforcement on the ban of CQ and that CQ is still in use after its official withdrawal in Kenya (*Rebelo et al., 2015*). The coastal region had the highest *Pfmdr* 1 86Y mutant allele with 23.4%. CQ selects for parasites with *Pfmdr* 1 86Y mutation (*Li et al., 2014*). Semi-Arid seasonal had the highest *Pfmdr* 1 184F mutant allele with 32.5% while the coastal endemic region had the highest *Pfmdr* 1 Y184 allele at 57.8% suggesting that the population in the Semi-Arid seasonal zone have a higher risk to parasite tolerance to some antimalarials. It has been reported that polymorphism in the *Pfmdr* 1 gene is associated with an increase in parasite tolerance/resistance to some anti-malarials (*Atroosh et al., 2016*). This would also imply increased use of lumefantrine in the region. It has been reported that existence of *Pfmdr* 1 184F mutant allele increases with the use of lumefantrine (*Malmberg et al., 2013*; *Sisowath et al., 2007*). An increase of 18.4% to 23.5% access to AL from within 48 h of fever on set had been reported in 2015 (*Wasunna et al., 2015*), after implementation of a new drug policy in Kenya 2004 (*Gitonga et al., 2008*; *Watsierah et al., 2012*). The high

parasitemia means that individuals living in the Lake endemic region will consequently have high rates of transmission of CQR genotypes as compared to other regions.

Previous studies show increasing chloroquine sensitivity (*Eyase et al., 2013*; *Kiarie et al., 2015*; *Mohammed et al., 2013*) after it was replaced by sulfadoxine-pyrimethamine in 1998 (*Eyase et al., 2013*). A study done in Tanzania reported >90% recovery of CQ susceptibility ten years after its withdrawal (*Mohammed et al., 2013*). Two studies done in Western Kenya reported a significant difference to CQ sensitivity in the *Pfcrt* 76 codon between 2008 and 2011 $p < 0.001$ (*Eyase et al., 2013*), and a decline from 76% to 6% of *Pfcrt* 76T prevalence from 2003 to 2015 (*Vardo-Zalik et al., 2018*). Another study done in Msabweni Kenya reported a 41% (n-99) in the *Pfcrt* 76T mutant allele in the year 2013 representing a significant decline in frequency compared to 2006 ($p \leq 0.05$) (*Kiarie et al., 2015*). Similarly, this study shows increasing frequency of *Pfcrt* K76 and *Pfmdr1* N86 genotypes that are suggestive of increasing CQ sensitivity in the four out of five malaria endemicity zones of Kenya heralding possible reintroduction of CQ based combination treatment. On the other hand, the prevalence *Pfmdr* 1 184F genotype increased during the 2008-14 period. The *Pfmdr* 1 184F mutation has been associated with AL resistance (*Antony et al., 2016*), and poses a threat to the sensitivity of lumefantrine and MQ (*Ministry of Health, 2010*). Lumefantrine in combination with artemether has been the recommended first-line treatment for uncomplicated malaria in Kenya since 2006 (*Gitonga et al., 2008*; *Lucchi et al., 2015*; *Ministry of Health, 2010*), while MQ is used as a chemoprophylaxis in Kenya (*Ministry of Health, 2010*).

Studies have shown that understanding the patterns of parasite dispersal from local hotspots of transmission can aid the design of additional targeted control by identifying both the regions where imported infections originate and where they may contribute substantially to transmission (*Wesolowski et al., 2012*). It will therefore be of great importance to establish gene flow patterns of *Plasmodium falciparum* resistant genotypes within the Kenyan population. This will be crucial in designing strategies towards control and prevention of malaria per epidemiological zone.

The limitation to this study was using a convenience sample. This implies that the sample size used in some regions could not be a representative of the whole population. Univariate and multivariate regression was used to analyze data that was stratified by age, gender and location to ensure that the analysis outcome is not due to confounding factors, bias or effect modifiers.

## CONCLUSIONS

According to this study gender and travel have contributed to the dispersion of CQR genotypes in Kenya. The coast and lake endemic region bears the brunt of the burden among other regions. Additionally, children under the age of 5 years remain to be at a higher risk of malaria burden. Decline in CQR based on *Pfcrt* 76T SNP, is an indication that CQ can be re-considered for treatment of *Pf* malaria in future. An increase in *Pfmdr* 1 N86 wild type implies tolerance to lumefantrine. On the other hand, increased *Pfmdr* 1 184F mutant genotype frequency indicates a reduced sensitivity to lumefantrine. This

study recommends continuous monitoring of CQR with wholesome information. Further studies should be done to determine migration patterns of *Pf* CQ resistant genotypes and gender factors associated with CQR in the malaria epidemic zones in Kenya.

## ACKNOWLEDGEMENTS

We thank all GEIS clinical staff serving at the sentinel sites for their assistance and the study subjects and their parents/guardians who participated in the study and contributed their samples for study.

### Funding

This work was supported by the U.S. Department of Defense Global Emerging Infections System, Silver Spring, Maryland. The funders had no role in study design, data collection and analysis, decision to publish, or preparation of the manuscript.

### Grant Disclosures

The following grant information was disclosed by the authors:
U.S. Department of Defense Global Emerging Infections System, Silver Spring, Maryland.

### Competing Interests

The authors declare there are no competing interests.

### Author Contributions

- Moureen Maraka conceived and designed the experiments, performed the experiments, analyzed the data, prepared figures and/or tables, authored or reviewed drafts of the paper, and approved the final draft.
- Hoseah M. Akala, Asito S. Amolo conceived and designed the experiments, analyzed the data, prepared figures and/or tables, authored or reviewed drafts of the paper, and approved the final draft.
- Dennis Juma, Duke Omariba, Agnes Cheruiyot, Benjamin Opot, Charles Okello Okudo, Edwin Mwakio, Gladys Chemwor, Jackline A. Juma, Raphael Okoth, Redemptah Yeda performed the experiments, analyzed the data, prepared figures and/or tables, authored or reviewed drafts of the paper, and approved the final draft.
- Ben Andagalu conceived and designed the experiments, analyzed the data, prepared figures and/or tables, authored or reviewed drafts of the paper, and approved the final draft.

### Human Ethics

The following information was supplied relating to ethical approvals (i.e., approving body and any reference numbers):

This study was approved by the Kenya Medical Research Institute (KEMRI) and Walter Reed Army Institute of Research (WRAIR) institutional review boards (protocol numbers: KEMRI 1330, WRAIR 1384, and HRPO Log #A-19306.3) respectively.

## Data Availability

The raw data is available in the Supplemental Files.

## Supplemental Information

Supplemental information for this article can be found online at http://dx.doi.org/10.7717/peerj.8082#supplemental-information.

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
