# Peer review of "A seven-year surveillance of epidemiology of malaria reveals travel and gender are the key drivers of dispersion of drug resistant genotypes in Kenya"

_PeerJ, doi:10.7717/peerj.8082_

## Round 0.1 · original submission · Major Revisions

The review process is now complete, and two thorough reviews from highly qualified referees are included at the bottom of this letter. All reviewers including myself agree the manuscript deserves to be published. Although there is considerable merit in your paper, we also identified some concerns that must be considered in your resubmission. Please, complete and correct the genotyping description of the molecular markers of resistance studied according to recommendations from the Reviewer #1. Please, improve the quality of Figure 5: since the names of the regions are unreadable, I suggest suppressing them in the three maps. Complete Figure 1 including an Africa Map depicting the studied area; indicate the scale of the maps. The authors also need to provide the coordinates of the studied areas in Material and Methods Section.

Reviewer 1 ·

Basic reporting

The article as it stands, is interesting but needs further improvement to achieve sufficient scientific relevance for publication. The paper describes the frequency of chloroquine resistance markers present in P. falciparum from distinct areas of Kenya throughout 7 years. This gives a broad view of the parasite selection happening in this region.
The resistant markers studied here, although authors emphasize its relevance for chloroquine, introduction should be changed for the importance of this markers in regards to the first line treatment used in Kenya throughout these 7 years. This work shows a clear selection of this markers and its relevance should be discussed towards the present treatment used in the region.
The PfCRT K76T SNP has been considered the major parasite risk factor for partner drug lumefantrine (Sisowath et al. 2009; Venkatesan et al. 2014) that the authors inform to be used in Kenya as one of the first line treatment (artemether-lumefantrine). Regarding PfMDR1 N86Y and Y184F, their impact on lumefantrine has been studied through gene editing strategies (Veiga et al. 2016). Its impact should be discussed considering its selection in Kenya through the years.
Moreover, PfCRT and PfMDR1 are both transporter proteins located in the food vacuole of the parasite. They are thought to regulate the flux of solutes within this membrane. Their functional relatedness is evidenced by the linkage disequilibrium observed between certain combination of alleles (Happi et al. 2006; Holmgren et al. 2006; Sisowath et al. 2009). With this in mind, the manuscript misses a haplotype analysis, through time, of the herein studied alleles.

What’s the second-line treatment, could this influence the selection of the alleles here in studied?

Suggestion: Copy number variation (CNV) of pfmdr1 has been linked to lumefantrine increased tolerance (Sidhu et al. 2006; Veiga et al. 2011). Although its prevalence in Africa is low, it will be interesting to study it. I would recommend at least to mention this type of polymorphism and its contribution to first-line treatment used in Kenya, in the discussion.


The discussion section should contain mentions to the figures and tables described in the manuscript.

Table 3 is not mentioned in any place of the manuscript.

Experimental design

Genotyping of the herein described molecular markers of resistance are not well described. Line 149: A fragment containing the PfCRT K76T SNP was amplified by nested PCR but there is no description of the genotyping discrimination. What was the method used to discriminate between amplified fragments with wildtype or mutant allele? RFLP, sanger sequencing…? What was the control used to discriminate between mutant and wildtype? Line 178: same problem for the markers present in PfMDR1. Authors describe the use of realtime PCR but Amplitaq Gold polymerase is not suitable for realtime PCR. How was the allele discrimination done? Realtime SNP genotyping (if this was the method, PCR conditions described are wrong) or RFLPs or sanger sequencing? What was the control used to discriminate between mutant and wildtype? The reference used to describe the methodology (Akala et al. 2011) is not correct since Akala et al 2011, refers to other papers for the allelic discrimination methodology used in their work.

Validity of the findings

"No comments"

Additional comments

Minor comments:

Line 64: “associated with single nucleotide polymorphisms in Pfcrt 76 gene”. Pfcrt 76 is not the name of the gene. Sentence should be rephrased. Suggestion: “associated with PfCRT K76T single nucleotide polymorphism.”
Line 67: “monitor possible confounding factors concurrently”. Describe some of the relevant confounding factors.

Line 105: “attending outpatient clinics from 2007 to 2014”. Results show data from 2008 to 2014.

Line 106: change “P f” to “P. falciparum”

Line 109: “2ml of blood was collected”… “Additionally, FTA filter paper was used to collect three blood spots”… . in this manuscript blood was only used for molecular analysis and therefore used only the blood collected into the FTA filter paper. For what they have used the 2mL collected blood?

Line 116: “100 ml each”. I believe author meant 100 µl.
Line 132: DNA extraction was done with QIAamp DNA Mini Kit. Therefore, I find it unnecessary to detail the extraction protocol (line 138-line 148) unless its differs from manufacture.

Line 148: “-200C until required.” The symbol used to describe the degrees Celsius is not correct. Please amend that throughout the paper.

Line 149: “Pfcrt 76 genotyping: Pfcrt genotyping was amplified by…”. Again the terminology is not correct. Suggestion: PfCRT K76T SNP analysis: A fragment of pfcrt gene spanning K76T SNP was amplified by…

Line 152: the reference paper for the genotyping condicions described (Lakshmanan et al. 2005), uses reverse transcription PCR followed by nested and sanger sequencing. This methodology might not be the one used in this manuscript once they extracted DNA and not RNA from clinical samples.

Line 169: “para film.”. amend to parafilm.

Line 174: “Negative and positive control was used”. Please describe the type of controls used.

Line 178: “Pfmdr1 86 and Pfmdr1 184 genotyping. Sequences of Pfmdr1 86 and Pfmdr1 184 genes were…” same comment as the one made for line 149. “of Pfmdr1 86 and Pfmdr1 184 genes” With this sentence it gives the impression the authors don’t know what is a gene and what is an allele!

Line 201: “clinical factors to chloroquine resistance.” Chloroquine resistance was not accessed in this manuscript. I believe the author meant to inform about the genotyping in the region, therefore, I suggest to complete with “clinical factors to chloroquine resistance molecular markers.”
Line 365: Parasite density discussed in this paragraph has been shown to fluctuate even in the very first hours of sample collection (Carlsson et al. 2011) alerting for caution when discussing its burden. Considering this manuscript with 2 figures (figure 2 and 3) on this subject, I suggest to add on the discussion this concern.

Line 415: SP needs detailed description.

Figure 5: the spatial distribution considers samples from the 7 years? If so, and considering the fluctuation of alleles prevalence shown in figure 4, to discriminate spatial distribution only for the recent year since average of all year does not give relevant information.


End of reviewer comments.




References used in the reviewer comments:
Akala HM, Eyase FL, Cheruiyot AC, Omondi AA, Ogutu BR, Waters NC, Johnson JD, Polhemus ME, Schnabel DC, and Walsh DS. 2011. Antimalarial drug sensitivity profile of western Kenya Plasmodium falciparum field isolates determined by a SYBR Green I in vitro assay and molecular analysis. Am J Trop Med Hyg 85:34-41. 10.4269/ajtmh.2011.10-0674
Carlsson AM, Ngasala BE, Dahlstrom S, Membi C, Veiga IM, Rombo L, Abdulla S, Premji Z, Gil JP, Bjorkman A, and Martensson A. 2011. Plasmodium falciparum population dynamics during the early phase of anti-malarial drug treatment in Tanzanian children with acute uncomplicated malaria. Malar J 10:380. 10.1186/1475-2875-10-380
Happi CT, Gbotosho GO, Folarin OA, Sowunmi A, Bolaji OM, Fateye BA, Kyle DE, Milhous W, Wirth DF, and Oduola AM. 2006. Linkage disequilibrium between two distinct loci in chromosomes 5 and 7 of Plasmodium falciparum and in vivo chloroquine resistance in Southwest Nigeria. Parasitol Res 100:141-148. 10.1007/s00436-006-0246-4
Holmgren G, Gil JP, Ferreira PM, Veiga MI, Obonyo CO, and Bjorkman A. 2006. Amodiaquine resistant Plasmodium falciparum malaria in vivo is associated with selection of pfcrt 76T and pfmdr1 86Y. Infect Genet Evol 6:309-314. 10.1016/j.meegid.2005.09.001
Lakshmanan V, Bray PG, Verdier-Pinard D, Johnson DJ, Horrocks P, Muhle RA, Alakpa GE, Hughes RH, Ward SA, Krogstad DJ, Sidhu AB, and Fidock DA. 2005. A critical role for PfCRT K76T in Plasmodium falciparum verapamil-reversible chloroquine resistance. EMBO J 24:2294-2305. 10.1038/sj.emboj.7600681
Sidhu AB, Uhlemann AC, Valderramos SG, Valderramos JC, Krishna S, and Fidock DA. 2006. Decreasing pfmdr1 copy number in plasmodium falciparum malaria heightens susceptibility to mefloquine, lumefantrine, halofantrine, quinine, and artemisinin. J Infect Dis 194:528-535. 10.1086/507115
Sisowath C, Petersen I, Veiga MI, Martensson A, Premji Z, Bjorkman A, Fidock DA, and Gil JP. 2009. In vivo selection of Plasmodium falciparum parasites carrying the chloroquine-susceptible pfcrt K76 allele after treatment with artemether-lumefantrine in Africa. J Infect Dis 199:750-757. 10.1086/596738
Veiga MI, Dhingra SK, Henrich PP, Straimer J, Gnadig N, Uhlemann AC, Martin RE, Lehane AM, and Fidock DA. 2016. Globally prevalent PfMDR1 mutations modulate Plasmodium falciparum susceptibility to artemisinin-based combination therapies. Nat Commun 7:11553. 10.1038/ncomms11553
Veiga MI, Ferreira PE, Jornhagen L, Malmberg M, Kone A, Schmidt BA, Petzold M, Bjorkman A, Nosten F, and Gil JP. 2011. Novel polymorphisms in Plasmodium falciparum ABC transporter genes are associated with major ACT antimalarial drug resistance. PLoS One 6:e20212. 10.1371/journal.pone.0020212
Venkatesan M, Gadalla NB, Stepniewska K, Dahal P, Nsanzabana C, Moriera C, Price RN, Martensson A, Rosenthal PJ, Dorsey G, Sutherland CJ, Guerin P, Davis TM, Menard D, Adam I, Ademowo G, Arze C, Baliraine FN, Berens-Riha N, Bjorkman A, Borrmann S, Checchi F, Desai M, Dhorda M, Djimde AA, El-Sayed BB, Eshetu T, Eyase F, Falade C, Faucher JF, Froberg G, Grivoyannis A, Hamour S, Houze S, Johnson J, Kamugisha E, Kariuki S, Kiechel JR, Kironde F, Kofoed PE, LeBras J, Malmberg M, Mwai L, Ngasala B, Nosten F, Nsobya SL, Nzila A, Oguike M, Otienoburu SD, Ogutu B, Ouedraogo JB, Piola P, Rombo L, Schramm B, Some AF, Thwing J, Ursing J, Wong RP, Zeynudin A, Zongo I, Plowe CV, Sibley CH, Group AMMS, and Wwarn AL. 2014. Polymorphisms in Plasmodium falciparum chloroquine resistance transporter and multidrug resistance 1 genes: parasite risk factors that affect treatment outcomes for P. falciparum malaria after artemether-lumefantrine and artesunate-amodiaquine. Am J Trop Med Hyg 91:833-843. 10.4269/ajtmh.14-0031

Reviewer 2 ·

Basic reporting

The manuscript addresses a very relevant question on the role of demographic factors for the spread of drug resistance and used comprehensive analyses to identify these factors. The manuscript is written very clearly and raw data is made available.

Experimental design

The study aims to establish the association of demographic factors with the spread of drug (at least chloroquine) resistance in Kenya. This manuscript presents a comprehensive analysis of factors that are associated with CQ resistance. The authors reported response to the question "Ever travelled in the last 2 months", however, the risk of travel is highly dependent on the destination (eg. malaria prevalence at the destination) and duration (eg. for how long they traveled? how likely are the travelers to get infected at the destination?).

Did the authors collect more information to characterize travel data? Can this be incorporated as a stratified analysis?

Validity of the findings

Minor comment:

1. Line 73: “These factors could consequently be associated with chloroquine resistance.” Do the authors refer to association with the dispersal of resistance? Please clarify

Major comments:

2. Line 219 – Is this log2 transformation? If the parasite density is reported as log10 (as stated in Figure 2 and 3) – Please clarify or correct accordingly. If the authors reported log10 values, the parasite density measured as “the number of parasites diagnosed by microscopy per microliter of whole blood” is extremely high.

In average human blood contains about ~ 5 million RBC per microliter of whole blood (notwithstanding anemia/malnutrition etc), therefore a parasite density of 5,000,000 (in log10=6.7) is almost 100% parasitemia. Although Pf can cause high parasitemia infections (rarely up to 20%) and certainly anything close to 100% (or more, in this case, log10 of 14.2) should be treated with caution. Based on the description provided, I suggest the authors clarify how the data was log-transformed

---

## Round 0.2 · accepted · Accept

All points raised by the reviewers and myself were properly addressed.

Reviewer 1 ·

Basic reporting

No Comment.

Experimental design

No comment.

Validity of the findings

No comment

Additional comments

Line 420: Type error "zones pf Kenya." pf - of
Line 425: "treatment (Carlsson et al., 2011), It is therefore" change the comma for a dot.
Line 448 - 450: The sentence "The implication of Pfmdr1 Y184F mutation has been reported to be minimal on antimalarial drug susceptibilities (Veiga et al., 2016)." should be taken with caution since this relates to specific genetic background and could lead the reader to misinterpret it. I would suggest to delete it since does not complement the discussion taken in this paragraph.